# The Binomial “Inflammation-Epigenetics” in Breast Cancer Progression and Bone Metastasis: IL-1β Actions Are Influenced by TET Inhibitor in MCF-7 Cell Line

**DOI:** 10.3390/ijms232315422

**Published:** 2022-12-06

**Authors:** Daniele Bellavia, Viviana Costa, Angela De Luca, Aurora Cordaro, Milena Fini, Gianluca Giavaresi, Fabio Caradonna, Lavinia Raimondi

**Affiliations:** 1SC Scienze e Tecnologie Chirurgiche (STC), IRCCS Istituto Ortopedico Rizzoli, 40136 Bologna, Italy; 2Direzione Scientifica, IRCCS Istituto Ortopedico Rizzoli, 40136 Bologna, Italy; 3Dipartimento di Scienze e Tecnologie Biologiche Chimiche e Farmaceutiche, Università degli studi di Palermo, Viale delle Scienze, Edificio 16, 90128 Palermo, Italy

**Keywords:** DNA methylation, bone metastasis, inflammation, Interleukin-1β, ten-eleven translocation proteins, MCF-7 cell line

## Abstract

The existence of a tight relationship between inflammation and epigenetics that in primary breast tumor cells can lead to tumor progression and the formation of bone metastases was investigated. It was highlighted how the induction of tumor progression and bone metastasis by Interleukin-1 beta, in a non-metastatic breast cancer cell line, MCF-7, was dependent on the de-methylating actions of ten-eleven translocation proteins (TETs). In fact, the inhibition of their activity by the Bobcat339 molecule, an inhibitor of TET enzymes, determined on the one hand, the modulation of the epithelial-mesenchymal transition process, and on the other hand, the reduction in the expression of markers of bone metastasis, indicating that the epigenetic action of TETs is a prerequisite for IL-1β-dependent tumor progression and bone metastasis formation.

## 1. Introduction

Breast cancer is the most frequent tumor in women worldwide and is considered curable only in the early stage, before metastasis development [1]. Advanced breast cancer develops often as bone metastasis, and this condition determines a life expectancy limited to 2 to 3 years after diagnosis [2]. Different molecules are involved in the intricate network of signaling pathways able to orchestrate communication between cancer cells and the surrounding stroma. The microenvironment is critical for promoting tumor progression and to attract metastasizing cancer cells to bone [3]. The most recent advances to understand tumor–microenvironment interactions have elucidated the crucial role of the immune system, in particular of the inflammatory process, in supporting or inhibiting tumor progression and metastasis [3]. In breast cancers, the alteration of DNA methylation is a prevalent epigenetic feature associated with an increased risk of metastasis and poor prognosis. During breast tumor progression, the cancer cell undergoes paradoxical changes in DNA methylation; regional hypermethylation silences genes involved in the cell cycle regulation, growth regulation, and pro-apoptotic genes [4]; and global hypomethylation is indicated as a prerequisite of tumor progression, epithelial-mesenchymal transition (EMT), and metastasis [5].

Inflammation is a dynamic endogenous process, and it is known that its deregulation could result in a variety of inflammation-related diseases. Emerging evidence indicates the very important role of epigenetic modifications including chromatin remodeling, DNA modifications, Histone modifications, and non-coding RNA (miRNAs, lncRNA, etc.), which are able to modulate multiple signaling pathways of chemokines and cytokines released by immune cells in the inflammation response [6,7]. Recent studies evidenced how the inflammatory process can lead to epigenetic modifications; DNA methylation, in particular, results in the activation or deactivation of genes related to tumor progression and metastasis [8]. In fact, DNA methylation modulation is implicated in various biological processes, including cell activation, proliferation, differentiation, and apoptosis. Its aberrant regulation is associated with tumor progression and metastasis in several tumor types [9]. In particular, Interleukin-1β (IL-1β) stimulation led to a strong remodulation of chromatin accessibility, in part determined by the remodulation of genome methylation that in tumor cells is often associated with tumor progression [10,11,12,13].

The current study aimed to elucidate direct and indirect evidence that linked inflammation to DNA methylation regulation. Notably, we found that Bobcat339, belonging to a new class of cytosine-based TET enzyme inhibitor, was able to inhibit reversibly the hydroxylation activity of these enzymes [14]. This inhibition has permitted as evidence the role of de-methylation of genomic DNA in IL-1β-induced tumor progression and bone metastasis in MCF-7 non-metastatic breast cancer cell lines. After the first evidence [15], the existence of the binomial “Inflammation-Epigenetics” has begun to also make its way in tumor progression and metastasis.

## 2. Results

### 2.1. Methylase and Demethylase Inhibitors Change Genomic DNA Methylation in MCF-7 Cell Lines Treated with IL-1β

We investigated how an inflammation status, here mimicked by inflammatory cytokine IL-1β treatment, affected the proliferation of the breast cancer cell line MCF-7 Figure 1A shows the viability of MCF-7 was increased by the IL-1β treatment both in concentration and over time (*p* < 0.0005).

Starting from our hypothesis that IL-1β acts at the level of the regulation of DNA methylation, we carried out a spot hybridization of DNA extracted by MCF-7 under treatment with IL-1β, AZA (10 µM), and Bobcat339 at the two different concentrations able to inhibit only TET-1 (33 µM) or both TET-1 and TET-2 (75 µM) [14], to evaluate the modification in the methylation status of the different treatments, using antibody anti-methyl-cytosine or anti-hydroxy-methyl-cytosine, the first product of modification of methyl-cytosine, operated by the TET enzyme. Figure 1B indicates that the IL-1β treatment induced a general de-methylation of genomic DNA, similarly to AZA treatment, while the Bobcat operated contrasting this action, as indicated by the difference in the methyl-cytosine (*p* < 0.005) and hydroxy-methyl-cytosine (*p* < 0.05) content of genomic DNA isolated by the different treatments with respect to untreated cells.

To investigate if the modulators of DNA methylation interfere or not with IL-1β-induced proliferation, we carried out a viability assay (MTT assay) of MCF-7 treated with IL-1β or the inhibitor of DNA methylation (AZA) and inhibitor of TETs (Bobcat), and their relative co-treatments (Figure 1C). The treatment with only AZA or both concentrations of BC (33 µM and 75 µM) did not interfere with proliferation with respect to the control (*NS*), while in co-treatment with IL-1β, we observed that both concentrations of BC were able to interfere with IL-1β-induced proliferation (*p* < 0.0005), contrarily to AZA/IL-1β co-treatment that showed differences with respect to the IL-1β treatment (Figure 1C).

For these reasons, we decided to use the AZA treatment as the epigenetic positive control, acting epigenetically on DNA methylation similarly to IL-1β, while Bobcat was used only in co-treatment with IL-1β, contrasting epigenetically with the activities of IL-1β.

We then evaluated the migration abilities of MCF-7 cells by performing wound healing under the different treatments of IL-1β, AZA, and the co-treatment of IL-1β and Bobcat339 (Figure 2). The results showed that IL-1β was able to increase the closure of the wound, with respect to the untreated cells (*p* < 0.0005). Regarding the AZA treatment, an increase in healing speed in comparison to the untreated cells (*p* < 0.0005) was also found here. Furthermore, Bobcat treatments (33 µM and 75 µM) were able to revert the accelerated wound closure of IL-1β (Figure 2).

### 2.2. IL-1β Regulates IL-6 and IL-8 Releases through DNA Methylation Modifications

It is known in the literature that IL-1β is able to induce ex-novo IL-6 and IL-8 gene expression in the MCF-7 cell line [16]. To investigate if IL-1β actions determine a change in methylation status of IL-6 and IL-8 promotors, we conducted an MSRE-PCR of six sites in the IL-6 promoter and in two sites in the IL-8 promotor, containing CpG islands. We highlighted that IL-1β was able to reduce the methylation status in all analyzed sites, with respect to the untreated controls (untreated; *p* < 0.0005). In parallel, the treatment with the de-methylation agent AZA reduced greatly the methylation of these promotors (*p* < 0.0005). The co-treatment of IL-1β with BC blocked the de-methylation process induced by IL-1β, in all sites of IL-6 and IL-8 promotors (Figure 3A–C,E). The ELISA assay showed that the IL-1β treatment was able to induce the secretion of IL-6 and IL-8, as well as the AZA treatment, while the Bobcat was able to block IL-1β-dependent IL-6 secretion and partially that of IL-8 (Figure 3D,F).

### 2.3. AZA and BC Affect EMT Process induced by IL-1β Treatment

The action of the inhibitors of enzymes that regulate DNA methylation (DNMTs and TETs) on the IL-1β-induced EMT process of MCF-7 breast cancer cell line was subsequently assessed. At first, a Western blotting analysis was conducted to compare the principal markers of the epithelium-mesenchyme transition process, N-cadherin, E-cadherin, and Vimentin, between IL-1β and AZA treatments with respect to untreated cells, which were the last reported to induce metastasis through their overall de-methylation. In fact, the AZA treatment induced modifications on marker expressions of the EMT process, similarly to the IL-1β treatment (Figure 4A). Then, the Western blotting analysis highlighted an increase in N-cadherin and Vimentin, and a decrease in epithelial marker E-cadherin, under the IL-1β treatment (*p* < 0.0005), while the treatment with BC at both concentrations blocked the EMT process in IL-1β/BC co-treatments (see Figure 4B).

### 2.4. AZA and BC Affect Expression of Factors Involved in Bone Homing and Metastasis in MCF-7 Treated with IL-1β

The expression of bone homing inductors of the circulating breast cancer cells, also identified as bone metastasis markers in breast cancer, such as Versican, Osteopontin, and Prolactin receptor [17,18,19,20,21,22,23], were investigated. Western blot results showed an increased expression of these proteins in the IL-1β treatment (*p* < 0.0005), while the AZA treatment increased the expression of Osteopontin and Versican (*p* < 0.0005); no significative differences in Prolactin receptor expression (Figure 5A) were observed. On the contrary, Bobcat339 actions blocked the increased expression of these proteins induced by IL-1β treatment (*p* < 0.0005) (Figure 5B).

Finally, we also evaluated the ability of the macrophage cell-like Raw264.7 cells to form bone-resorbing OCs on dentine slices when cultured with the conditioned media (CM) from MCF-7 cells under the different conditions (Figure 6). Notably, the effect induced by Bobcat339 not only counteracted the increased invasiveness of MCF-7 cells, induced by the inflammatory environment, but would seem to restore MCF-7 cells to an initial condition, despite co-treatment with IL-1β, reducing the number of the resorption lacunae (evidenced as dark spots; Figure 6A).

### 2.5. IL-1β Treatment Induces Modification in DNMT and TET Expressions

We carried out a Western blot analysis using antibody anti-DNMT and TET enzymes. The treatment of IL-1β, such as AZA, increased TET-1, DNMT1, and DNMT3B protein expressions; IL-1β and Bobcat339 co-treatment did not change the result observed since Bobcat339 acted exclusively on the TET activity, independently to Bobcat339 co-treatment; the TET-2 protein expression resulted in no modification. Differently, DNMT3A showed a modulation of its expression related to Bobcat339 treatment (see Figure 7).

## 3. Discussion

IL-1β is the principal inflammatory cytokine induced and activated following infections or tissue damage. High levels of IL-1β in the tumor microenvironment are associated with high proliferation and aggressive phenotype [2]. IL-1β is able to induce the progression of breast cancer from a benign to a malignant stage, resulting in the deregulation of several growth factors, cytokines, and chemokines [24]. However, the mechanisms underlying these deregulated processes are still poorly understood.

Data obtained from this study demonstrated that IL-1β had the capacity to induce changes in gene expression, and modulate genome DNA methylation status, resulting in it being a prerequisite for tumor progression and bone metastasis. This modulation seems tightly related to the DNA de-methylation of the MCF-7 cell line genome, indicated by a reduction in methyl-cytosine and an increase in hydroxy-methyl-cytosine genomic contents, similarly to the AZA treatment (Figure 1B). AZA is a de-methylation agent that binds DNMT enzymes, blocking their activities, and in some tumors is indicated as a chemotherapeutic agent. Nevertheless, AZA is suspected to induce metastasis [25,26] through the same general de-methylation that sensitizes it to apoptosis [27]. We hypothesized that Bobcat339, acting epigenetically against de-methylation (Figure 1B), blocked IL-1β-dependent epigenetic actions, interfering with EMT and bone metastasis.

The MTT assay showed IL-1β induced an increased proliferation of the MCF-7 cell line, unlike the AZA and Bobcat treatments. Interestingly, only the co-treatment with Bobcat, and not with AZA, was able to revert IL-1β-induced proliferation, indicating that this action was partially related to TET activity blocked by Bobcat339. Furthermore, the wound healing assay evidenced that as IL-1β increased the migration ability of MCF-7, Bobcat339 was able to reduce the IL-1β-induced accelerated closure (Figure 2). We also observed that AZA was able to accelerate wound closure. This was justified by the evidence that the AZA treatment could induce an increased motility of the cells [25], confirmed by the evidence of the increased EMT process in the AZA treatment, indicated by the EMT markers (Figure 4).

It is known that non-metastatic breast cancer cells do not express IL-6 and IL-8; our data showed that treatment with IL-1β induced their expressions, through the de-methylation of their promotors (Figure 3A–C,F). IL-6 and IL-8 expression are described as the first step of tumor progression and metastasis in breast cancer, and their levels are related to the disease stage of breast cancer patients [28,29,30]. The Bobcat339 treatment blocked the promotor de-methylation of IL-6 and IL-8 genes, and consequently their protein expression, indicating a direct action of TET proteins on these genes (Figure 3D,F).

Similarly, Bobcat339 was able to block the EMT process, directly or indirectly, as indicated by the EMT markers in Western blots (Figure 4). Furthermore, the increased expression of bone metastasis markers (Versican, Osteopontin, and Prolactin receptor) induced by IL-1β- and AZA treatments, and not observed in IL-1β/Bobcat co-treatments (Figure 5), confirmed DNA de-methylation as a prerequisite for bone metastasis. Versican is a proteoglycan that is an essential structural factor of the extracellular matrix overexpressed in aggressive breast cancer [17]. Its high expression leads to the enhancement of breast cancer progression, resistance to chemotherapy, and an enhanced ability to metastasize to bone [18,23]. Osteopontin represents a crucial mediator of cellular cross talk and a key factor in the tumor microenvironment. In breast cancer, a high expression of Osteopontin was associated with frequent osteolysis, inducing the expression of bone-resorbing proteases, cathepsin K, and MMP9. Furthermore, the expression of this glycoprotein in breast cancer cells is reported to play a role in the bone homing of circulating breast metastatic cells [19,20]. The Prolactin receptor is involved in a wide range of cellular processes from physiological protective actions to pathological activities such as bone metastasis formation and accelerated osteolytic lesions. In addition, the expression of the Prolactin receptor in primary breast cancer is associated with poor prognosis, due to accelerated progression and bone osteolytic metastasis development [21,22].

The ability of the conditioned medium of IL-1β-treated MCF-7 cells to induce OC differentiation and activity through the bone resorption pit assay, despite its inability in the co-treatment with Bobcat339, indicated a potential involvement of epigenetic activity of IL-1β in the induction of the bone osteolytic mechanism of breast cancer.

The involvement of the enzymes implicated in the modulation of DNA methylation was shown in the Western blot analysis (Figure 7), evidencing how IL-1β treatment was able to induce TET-1, DNMT1, and DNMT3B expression, independently by the Bobcat co-treatment. Only DNMT3A expression seemed to be regulated by the Bobcat339 treatment, being downregulated by the IL-1β treatment, and upregulated by the IL-1β/Bobcat339 treatment. It is known that increased DNMT1 and DNMT3B expression play a significant role in the development and progression of several cancers, increasing the promotor methylation of several tumor suppressor genes [31,32,33,34]. The modulation of the EMT process and the reduction of bone metastasis markers and bone resorption activity in the IL-1β/Bobcat339 co-treatments indicated that the IL-1β actions were linked to the activity of TET enzymes.

The pro-inflammatory process carried out by IL-1β is known to promote TET-1 expression, which in turn regulates pro-inflammatory cytokines [35,36]. The current results suggest TET-1 as responsible for the IL-1β-dependent tumor progression of the MCF-7 cell line, indicating its potential therapeutic treatment as a druggable target in breast cancer acting on progression and bone metastasis.

Further studies are needed to confirm the molecular mechanisms that identify IL-1β-dependent TET-1 overexpression as a prerequisite for tumor progression and bone metastasis.

## 4. Materials and Methods

### 4.1. Reagents

Dulbecco’s modified Eagle’s high glucose medium (DMEM) was purchased from the Lonza Group (Basel, Switzerland); fetal bovine serum (FBS), L-glutamine, and penicillin/streptomycin were purchased from the Lonza Group. Azacytidine (AZA) was purchased from Sigma-Aldrich (Merck KGaA group, Darmstadt, Germany). Bobcat339 hydrochloride (BC) was purchased from MedChemExpress (MCE, Monmouth Junction, NJ, USA).

### 4.2. Cell Cultures

The breast cancer cell line MCF-7 (HTB-22™), purchased from ATCC^®^ (Manassas, VA, USA), was cultured at 37 °C and 5% CO_2_ in DMEM (Euroclone S.p.A., Pero, Milan, Italy) supplemented with 10% heat-inactivated FBS (Lonza, Verviers, Belgium), 1 mM Sodium Pyruvate (Euroclone), 2 mM glutamine, 100 U/mL penicillin, and 100 μg/mL streptomycin (Gibco, Invitrogen Corp., Carlsbad, CA, USA).

### 4.3. Viability Assay (MTT Assay)

Cell viability was assessed with the Methyl-thiazoltetrazolium (MTT) assay as previously described [37]. Briefly, MCF-7 cells were seeded at a density of 1 × 10^5^ in a 96-well plate and exposed to 12.5 ng/mL and 25 ng/mL of IL-1β for 24, 48, and 72 h. Similarly, we measured the viability of MCF-7 cells also exposed to the final concentration of 10 µM of AZA, and 33 µM and 75 µM of Bobcat339, relative to the co-treatment with IL-1β, for 24, 48, and 72 h. The absorbance was measured at 540 nm.

### 4.4. Genomic DNA (gDNA) Extraction

The isolation of the genomic DNA of the MCF-7 cell line, under different treatments, was carried out with the PureLink Genomic DNA mini-kit (Invitrogen™, Waltham, MA, USA). The DNA was quantified using the Nanodrop 2000 spectrophotometer (ThermoFisher Scientific, Waltham, MA, USA). The integrity of DNA was successively analyzed in 0.8% agarose gel electrophoresis and visualized through Gel Red staining (Biotium, Hayward, CA, USA) in a ChemiDoc apparatus (Bio-Rad Laboratories, Hercules, CA, USA).

### 4.5. Dot Spot Hybridization Analysis

Isolated gDNA (1 µg) were spotted in the Hybond N+ membrane as described elsewhere [38]. Denatured DNAs were fixed in the membranes with U.V. treatment for 2 min. The membranes were soaked in 5% BSA in TBS-T for 1 h, to block non-specific antibody binding sites. After washing for 5 min three times in TBST, the membranes were incubated with a rabbit anti-5-methylcytosine (5-mC) (Invitrogen™, 1:1000 dilution) or rabbit anti-5-hydroxy methyl-cytosine (5-hmC) (Invitrogen™, 1:1000 dilution) monoclonal antibody in TBS-T at 4 °C overnight. After three washes of the membrane, for 5 min in TBS-T, they were incubated with secondary anti-rabbit IgG HRP-linked antibody (1:2000, #7074—Cell Signaling) in TBS-T for 1 h at room temperature. The membranes were washed for 5 min three times in TBS-T. The enzyme substrate was added to the membrane. The secondary antibody signals were visualized using a chemiluminescence kit (Novex™ ECL Chemiluminescent Substrate Reagent Kit, Invitrogen™), through a ChemiDoc apparatus (Bio-Rad Laboratories, Hercules, CA, USA).

### 4.6. Wound Healing Assay

MCF-7 cells were cultured into 6-well plates. When the cell confluence reached 90%, a 10 µL pipette tip was used to scrape across the confluent cell layer, followed by gentle washing with phosphate-buffered saline (PBS), and the addition of DMEM high glucose supplemented with 10% FBS. Images were captured at 0, 6, 12, and 24 h using a Nikon Eclipse Ti microscope (Nikon Europe B.V., Amsterdam, The Netherlands). The wound healing ability under different treatments was determined by measuring the alteration of the distance of the two edges of the wound due to cell proliferation/migration, using NIH Image J software (NIH, Bethesda, MD, USA).

### 4.7. MSRE–PCR Analysis

A methylation-sensitive restriction endonuclease–PCR (MSRE–PCR) analysis was performed to determine the methylation status of the CpG-rich sites, present in the proximal promotor region of Interleukin-6 (IL-6) and Interleukin-8 (IL-8). Experiments were carried out as described elsewhere [15,39,40]. PCR products were analyzed by 2% agarose gel electrophoresis visualized by Gel Red staining (Biotium, Hayward, CA, USA) in a ChemiDoc apparatus (Bio-Rad Laboratories, Hercules, CA, USA), and the image captured in the digital support and densitometric analysis was obtained using the “Image Lab” application (version 5.2.1) of Bio-Rad Laboratories (Hercules, CA, USA).

### 4.8. ELISA Assay

In total, 2 × 10^4^ MCF-7 cells were plated on a 96-well plate and allowed to adhere overnight. The medium was then replaced, and cells were permitted to grow for 24 and 48 h. Cells were treated with IL-1β (25 ng/mL), AZA (10 µM), or two concentrations of Bobcat (33 µM and 75 µM), or co-treatment with IL-1β (25 ng/mL) in the presence of AZA (10 µM), or two concentrations of Bobcat (33 µM and 75 µM) for 24 and 48 h. The levels of secreted IL-6 or IL-8 were measured in culture supernatants by an enzyme-linked immunosorbent assay (ELISA) kit (R&D Systems Europe, Ltd., Abingdon Science Park, Abingdon, UK) according to the manufacturer’s instructions.

### 4.9. Western Blot Analysis

The SDS-PAGE electrophoresis and Western blotting were performed using cells lysed for 1 h in NP40 Cell lysis buffer containing 50 mM Tris, pH 7.4, 250 mM NaCl, 5 mM EDTA, 50 mM NaF, 1 mM Na3VO4, 1% Nonidet P40 (NP40), and 0.02% NaN3 (Invitrogen™, Thermo Fisher Scientific, Waltham, MA, USA); 1 mM PMSF (1M, Sigma–Aldrich, St. Louis, MO, USA) and Protease Inhibitor Cocktail (100X, Sigma–Aldrich, St. Louis, MO, USA) were added to the cell lysis buffer. To separate the cell lysates (30 µg per lane), we used the 4–12% Novex Bis-Tris SDS-acrylamide gels (Invitrogen™, Thermo Fisher Scientific, Waltham, MA, USA), transferred on Nitrocellulose membranes (Invitrogen™, Thermo Fisher Scientific, Waltham, MA, USA) through the iBlot 2 Dry Blotting System (Invitrogen™, Thermo Fisher Scientific, Waltham, MA, USA), and immunoblotted with the primary antibodies. The following antibodies against the following proteins were used: GADPH (1:1000, sc-5298, Santa Cruz Biotechnology, Inc., Dallas, TX, USA), Vimentin, E-cadherin (E-cad), N-cadherin (N-cad) (Epithelial-Mesenchymal Transition (EMT) Antibody Sampler Kit, #9782, Cell Signaling Technology, Danvers, MA, USA), Versican (Vcan) (1:1000, s351-23, Stressmarq Bioscience, Victoria, BC V8N, Canada), Osteopontin (Opn) (1:1000, NBP1-59190, Novusbio, Centennial, CO, USA), Prolactin receptor (Prlr) (1:1000, #13552, Cell Signaling Technology, Danvers, MA, USA), DNMT1 (1:1000, sc-271729, Santa Cruz Biotechnology, Inc., Dallas, TX, USA), DNMT3A (1:1000, sc-365769, Santa Cruz Biotechnology, Inc., Dallas, TX, USA), DNMT3B (1:1000, sc-376043, Santa Cruz Biotechnology, Inc., Dallas, TX, USA), ten-eleven translocation protein (TET)-1 (1:500, GT-1462, Invitrogen™, Thermo Fisher Scientific, Waltham, MA, USA), and TET-2 (1:500, CL-6873, Invitrogen™, Thermo Fisher Scientific, Waltham, MA, USA), and the secondary anti-mouse IgG HRP-linked antibody (1:2000, #7076, Cell Signaling Technology, Danvers, MA, USA) and anti-rabbit IgG HRP-linked antibody (1:2000, #7074, Cell Signaling Technology, Danvers, MA, USA).

### 4.10. Bone Resorption Pit Assay

A bone resorption assay was carried out as previously described [41], with little differences. Raw264.7 cells were seeded at a density of 5 × 10^4^ cells/mL in 96-well plates on organic dentine discs (Pantec, Torino, Italy) and cultured with either hrRANKL (25 ng/mL) or conditioned medium (DMEM high glucose medium supplemented with 10% FBS) of untreated MCF-7, AZA (10 µM) treated MCF-7, IL-1 β (25 ng/mL) treated MCF-7, and IL-1β (25 ng/mL)/Bobcat339 (33 µM) co-treated MCF-7cells.

The dentine discs, after 5 days of culture, were rinsed with 70% sodium hypochlorite for 5 min and fixed in 4% glutaraldehyde for 3 min. The resorption pits were stained using 1% toluidine blue and observed with a light microscope (Leica DM2500 Microsystems, Wetzlar, Germany) at a ×10 magnification. Three fields of each dentin disc for each experimental point were scored in three independent experiments. The number of the pits was calculated by NIH imageJ software analysis (plugin downloaded on https://rsbweb.nih.gov/ij/, accessed on 05 December 2022) [42].

### 4.11. Statistical Analysis

The statistical analysis was performed by using R software v.4.2.1 [43] and specific packages. One- or two-way ANOVAs were used to evaluate significant effects and/or interactions of selected factors (‘treatment’ for one- and two-way, and ‘experimental time’ for two-way) on normally distributed (Shapiro–Wilk test) data with homogeneity of variance (Levene test). Then, selected pairwise multiple comparisons with *p*-value adjusted according to the Sidak–Holm method were analyzed.

## 5. Conclusions

In conclusion, the current findings provide knowledge on the significant role of epigenetic actions of IL-1β on tumor progression, bone homing, and consequently bone metastasis formation. The block of TET actions through Bobcat339, understood as its epigenetic action, might be pivotal to the EMT process and bone metastatization (see schematic representation in Figure 8). Data obtained provided novel evidence on the implication of the epigenetic modification of the pro-inflammatory process on the enhanced motility of breast cancer cells and bone metastasis induction.

## Figures and Tables

**Figure 1 ijms-23-15422-f001:**
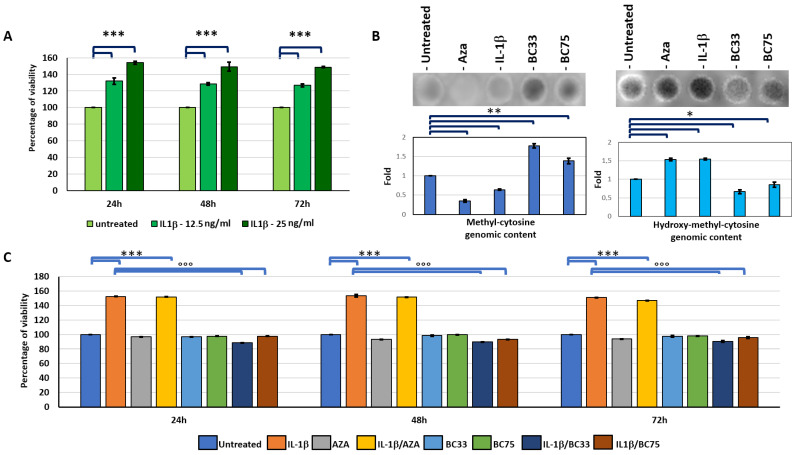
(**A**) Percentage of viability of MCF-7 (MTT assay) treated with 2 different concentrations of IL-1β (12.5 and 25 ng/mL) at 24, 48, and 72 h (Mean ± SD, *n* = 4). Two-way ANOVA showed a significant interaction (*p* < 0.0005) of treatment and experimental time factors on MCF-7 viability results. Significant differences among tested treatments at each experimental time (***) are reported in the graphs (1 symbol, *p* < 0.05; 2 symbols, *p* < 0.005; 3 symbols, *p* < 0.0005). (**B**) Representative images and diagram of densitometric analysis of spot hybridization of DNA from cell MCF-7 untreated or treated with AZA (10 µM), IL-1β (25 ng/mL), BC33 (Bobcat339, 33 µM), and BC75 (Bobcat339, 75 µM) using antibody through methyl-cytosine and hydroxy-methyl-cytosine (Mean ± SD, *n* = 3). One-way ANOVA showed the significant effect of treatment on densitometric data (*p* < 0.0005); significant differences among tested treatments vs. untreated MCF-7 are reported in the graphs (1 symbol, *p* < 0.05; 2 symbols, *p* < 0.005; 3 symbols, *p* < 0.0005). (**C**) Percentage of viability of MCF-7 treated with IL-1β, AZA, BC33, BC75, and co-treatment IL-1β/AZA and IL-1β/BC33 or BC75 (Mean ± SD, *n* = 4). Two-way ANOVA showed a significant interaction (*p* < 0.0005) of treatment and experimental time factors on MCF-7 viability results. Significant differences among tested treatments vs. untreated MCF-7 (*), and vs. IL-1β MCF-7 (°) are reported in the graphs ((*, *p* < 0.05; ** *p* < 0.005; *** and °°°, *p* < 0.0005).

**Figure 2 ijms-23-15422-f002:**
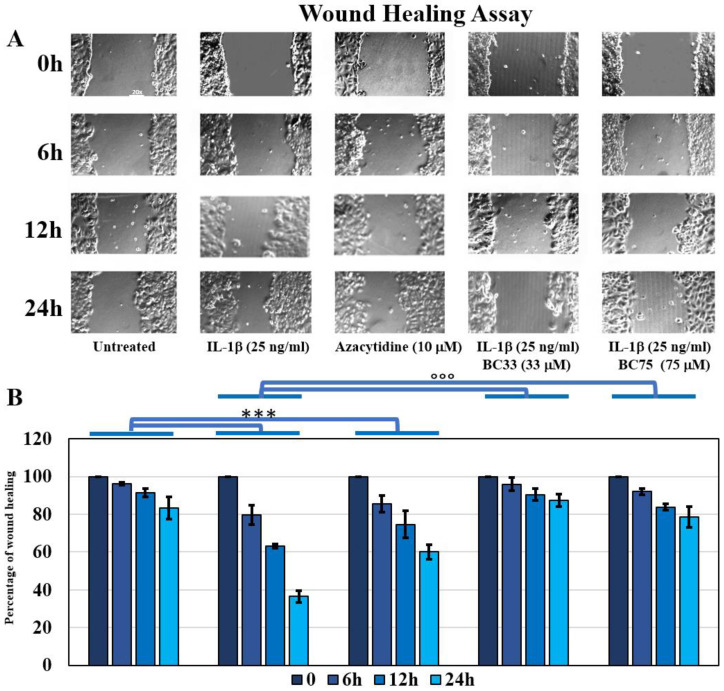
(**A**) Representative images of wound healing assay of MCF-7 cell line following the different treatments (20× magnification). (**B**) Percentage of wound healing for each tested treatments at each experimental time (Mean ± SD, *n* = 3). Two-way ANOVA showed a significant interaction of treatment and experimental time factors on wound healing data (F = 38.70, *p* < 0.0005). Significant differences among tested treatments versus untreated MCF-7 (*), and versus IL-1β MCF-7 (°) are reported in the graphs (*** and °°°, *p* < 0.0005).

**Figure 3 ijms-23-15422-f003:**
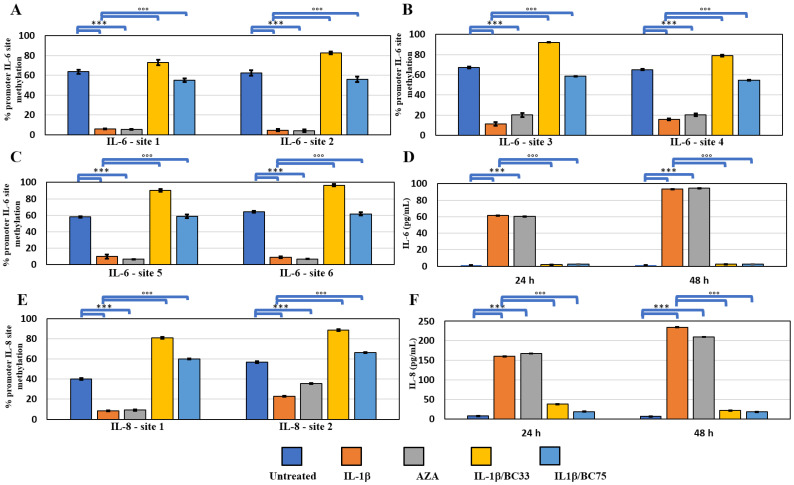
Methylation analysis of IL-6 and IL-8 promotors and relative releases of these cytokines. (**A**–**C**) Percentage of methylation of six methyl-sensible restriction sites through MSRE-PCR analyses of IL-6 promoter and (**D**) ELISA of secreted IL-6 by MCF-7 cell line after 24 and 48 h submitted to the following treatments: untreated; AZA(10 µM); IL-1β (25 ng/mL); IL-1β/BC33 (IL-1, 25 ng/mL; Bobcat339, 33 µM); IL-1β/BC75 (IL-1β, 25 ng/mL; Bobcat339, 75 µM) (Mean ± SD, *n* = 3, duplicates). One-way ANOVA highlighted significant effect (*p* < 0.0005) of treatment factor on percentage of IL-6 methylation sites, and two-way ANOVA a significant interaction (*p* < 0.0005) of treatment and experimental time factors on IL-6 release. (**E**) Percentage of methylation of two methyl-sensible restriction sites through MSRE-PCR analyses of IL-8 promoter and (**F**) ELISA of secreted IL-8 by MCF-7 cell line after 24 and 48 h under previously described treatments (Mean ± SD, *n* = 3, duplicates). One-way ANOVA highlighted significant effect (*p* < 0.0005) of treatment factor on percentage of IL-8 methylation sites and two-way ANOVA a significant interaction (*p* < 0.0005) of treatment and experimental time factors on IL-8 release. Significant differences among tested treatments versus untreated MCF-7 (*), and versus IL-1β MCF-7 (°) are reported in the graphs (*** and °°°, *p* < 0.0005).

**Figure 4 ijms-23-15422-f004:**
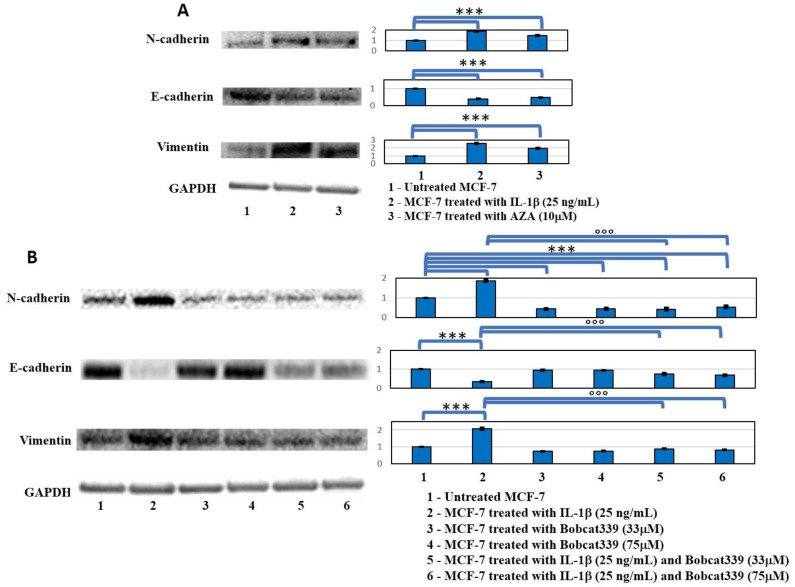
Western blot analysis of EMT markers. (**A**) Western blot analysis of EMT markers in MCF-7 cells untreated and treated with IL-1β and AZA after 48 h. Representative images of WB of N-cad, Vimentin, and E-cad, and relative densitometric analysis (Mean ± SD). One-way ANOVA showed significant effect of treatment on densitometric data (*p* < 0.0005); significant differences were highlighted among tested treatments (*p* < 0.0005). (**B**) Western blot analysis of markers that were modified in EMT process in MCF-7 cells untreated and treated with IL-1β and Bobcat339 (33 µM and 75 µM) and relative co-treatments after 48 h. Representative images of WB of N-cadherin, Vimentin, and E-cadherin, and relative densitometric analysis (Mean ± SD, *n* = 3). One-way ANOVA showed significant effect of treatment on densitometric data (*p* < 0.0005); significant differences among tested treatments versus untreated MCF-7 (*), and among IL-1β/BC co-treatments versus the IL-1β one (°) are reported in the graphs (*** and °°°, *p* < 0.0005).

**Figure 5 ijms-23-15422-f005:**
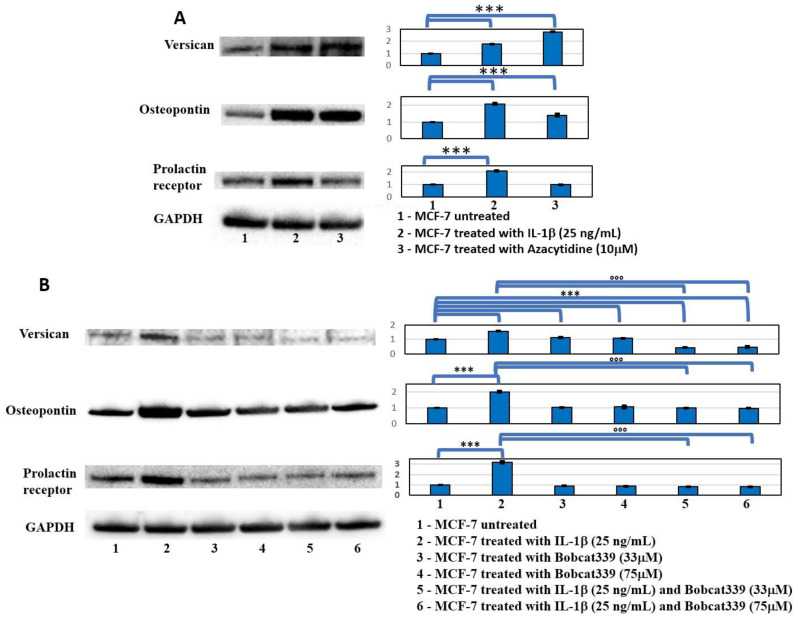
Western blot analysis of principal marker of bone homing. (**A**) Western blot analysis of Versican, Osteopontin, and Prolactin receptor in MCF-7 cells untreated and treated with IL-1β and AZA after 48 h. Representative images of WB and densitometric analysis (Mean ± SD) of Versican, Osteopontin, and Prolactin receptor. One-way ANOVA showed significant effect of treatment on densitometric data (*p* < 0.0005); significant differences were highlighted among tested treatments (*p* < 0.0005), except for Prolactin receptor with AZA (*NS*). (**B**) Western blot analysis of Versican, Osteopontin, and Prolactin receptor in MCF-7 cells untreated and treated with IL-1β and Bobcat339 (33 µM and 75 µM) and relative co-treatments after 48 h. Representative images of WB and densitometric analysis (Mean ± SD, *n* = 3) of Versican, Osteopontin, and Prolactin receptor. One-way ANOVA showed significant effect of treatment on densitometric data; significant differences among tested treatments versus untreated MCF-7 (*), and among IL-1β/BC co-treatments versus the IL-1β one (°) are reported in the graphs (*** and °°°, *p* < 0.0005).

**Figure 6 ijms-23-15422-f006:**
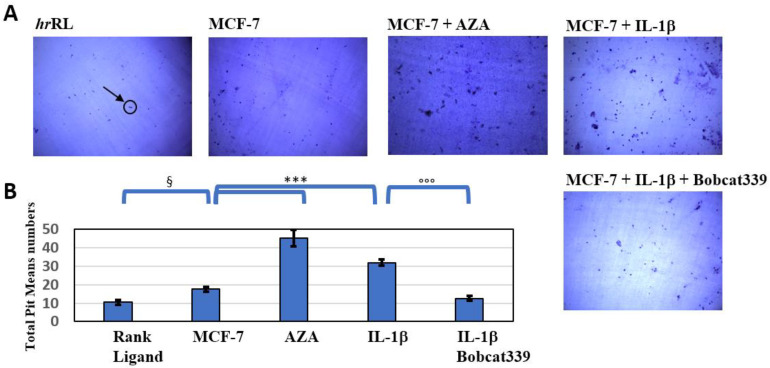
Bobcat339 hampers formation of lacunae on dentine slices. Bone resorption pit assay of Raw264.7 cells seeded on dentine discs and treated for 5 days with 25 ng/mL of hrRank Ligand or with the conditioned medium from MCF-7 cells (untreated or treated with IL-1β, AZA, or IL-1β/Bobcat339). (**A**) Representative images of lacunae formed (dark areas, as indicated by the arrow). (**B**) Data relative to the mean numbers of total pits formed by mature osteoclasts. One-way ANOVA showed significant effect of treatment on number of pit measured on dentins; significant differences among tested treatments versus hrRL (§), untreated MCF-7 (*), and among IL-1β/BC co-treatments versus the IL-1β one (°) are reported in the graphs (§, *p* < 0.05; *** and °°°, *p* < 0.0005).

**Figure 7 ijms-23-15422-f007:**
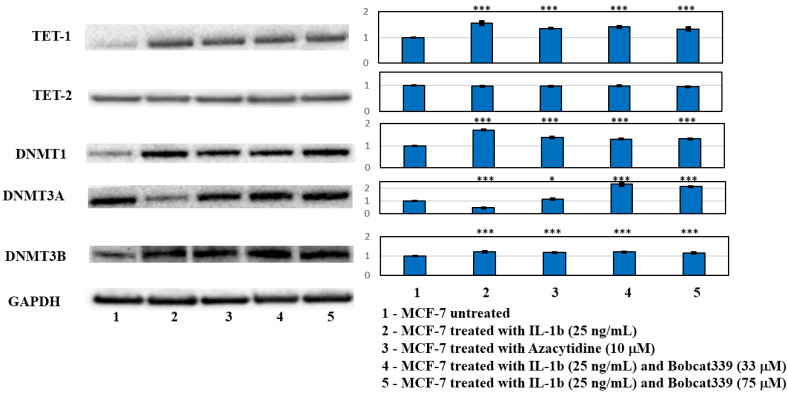
Western blot analysis of principal actors of DNA methylation process in MCF-7 cells untreated and treated with IL-1β, AZA (10 µM), and BC at the two concentrations (33 µM and 75 µM) and relative co-treatments after 48 h. Representative images of WB and densitometric analysis (Mean ± SD, *n* = 3) of TET-1, TET-2, DNMT1, DNMT3A, and DNMT3B. One-way ANOVA showed significant effect of treatment on densitometric data (*p* < 0.0005); significant differences were highlighted among tested treatments versus untreated MCF-7 (*p* < 0.0005), except for DNMT3A between AZA and untreated MCF-7 (*p* < 0.05). Furthermore, significant differences were found among IL-1β/BC co-treatments versus the IL-1β one (*p* < 0.0005), except for DBNT3B in IL-1β/Bobcat339 (33 µM) treatment. One-way ANOVA showed significant effect of treatment on densitometric data; significant differences among tested treatments versus untreated MCF-7 (*) are reported in the graphs (*, *p* < 0.05; ***, *p* < 0.0005).

**Figure 8 ijms-23-15422-f008:**
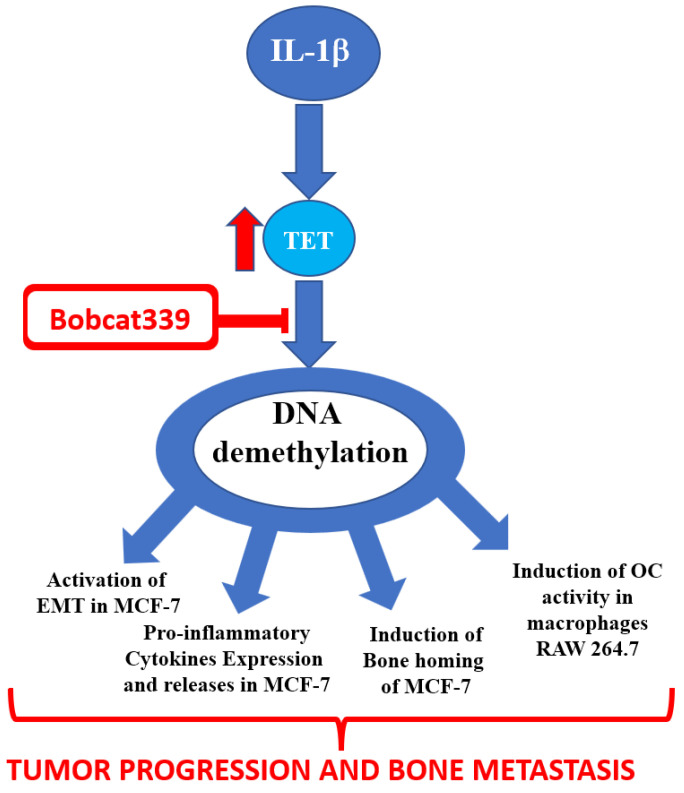
Schematic drawing that indicates Bobcat339 interferes in tumor progression and bone metastasis through TET inhibition.

## Data Availability

The data presented in this study are available on request from the corresponding author.

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
