# Peer review of "The Binomial “Inflammation-Epigenetics” in Breast Cancer Progression and Bone Metastasis: IL-1β Actions Are Influenced by TET Inhibitor in MCF-7 Cell Line"

_ijms, 2022, doi:10.3390/ijms232315422_

Round 1
Reviewer 1 Report
You have compelling evidence that bobcat33 reverses the actions of IL1b, although I am unconvinced this is through TET-1. The western blot is not showing a decrease in TET-1 when IL1b and bobcat 33 are given simultaneously. Can you show this another way - maybe gene expression is altered.
I don't think you have enough evidence to suggest TET-1 as the dependant of IL1b, you would need to show this through overexpression of TET-1 and the effect with IL1b on the assays you have performed.
More experiments needed or I suggest a change of title.
If you are showing that bobcat33 reverses methylation of IL1b, could this combined treatment be shown in figure 1B alongside the individual treatments.
If AZA is a positive control as mentioned, could this be included in the bone resorption assays
Author Response
Manuscript Number: ijms-2040827
Article Title: The binomial “Inflammation-Epigenetics” in breast cancer progression and bone metastasis: IL-1b actions are influenced by TET inhibitor in MCF-7 cell line.
Journal Name: International Journal of Molecular Sciences
reviewer’s requests
author’s answers
Reviewer #1 Comments
- You have compelling evidence that bobcat33 reverses the actions of IL1b, although I am unconvinced this is through TET-1. The western blot is not showing a decrease in TET-1 when IL1b and bobcat 33 are given simultaneously.
- We thank Reviewer#1 for the comments. Probably I should have given more information on Bobcat 339 actions. This molecule is an analogue of methyl-cytosine that bind TET enzymes blocking their actions, without modify TET-1 expression. In our work, we showed as IL-1b treatment induced an increase of expression of TET-1 enzyme. In co-treatment IL-1b/Bobcat339, IL-1b increase TET-1 expression (western blot analysis), while bobcat339 was able to block only its action (TET-1 expression was the same). We improved the text including this information.
- Can you show this another way - maybe gene expression is altered.
- For the reasons reported above, the expression of TET-1 in IL-1b treatment and relative co-treatments with bobcat339 does not change as bobcat acts only on TET-1 activity (therefore the levels of both the mRNA and TET-1 protein do not change).
- I don't think you have enough evidence to suggest TET-1 as the dependant of IL1b, you would need to show this through overexpression of TET-1 and the effect with IL1b on the assays you have performed. More experiments needed or I suggest a change of title.
- Indeed, further studies should be done to confirm the central role of TET-1 in the induction of tumor progression and in the formation of bone metastasis in MCF-7 cell line, therefore we decided to follow the reviewer's suggestion and change the title: The binomial “Inflammation-Epigenetics” in breast cancer progression and bone metastasis: IL-1b actions are influenced by TET inhibitor in MCF-7 cell line.
- If you are showing that bobcat33 reverses methylation of IL1b, could this combined treatment be shown in figure 1B alongside the individual treatments.
- This experiment has been carried out to see the trend of hydroxy-methyl cytosine and methyl-cytosine in genome of MCF-7 cells under the different treatment (IL-1b, AZA, and Bobcat339). This experiment highlighted as IL-1b determined a general demethylation of genome as AZA, while bobcat339 acted to the contrary. At this stage, in our opinion, it is premature to show these results of hydroxy-methyl-cytosine and methyl-cytosine levels in the genome of MCF-7 cells under bobcat/IL-1b co-treatments. However, we have carried out the Dot-spot hybridization both on genome of MCF-7 cells treated with Bobcat339 (the two concentrations) and relative IL-1b/bobcat339 co-treatments as requested. We did not observe significant differences on signals in co-treatment respect to bobcat339 treatment in spot hybridization using antibody through hydroxy-methyl-cytosine (first metabolite of TET that hydroxylates the methyl-cytosine), indicating the inhibition of TET activities. Differently, the results of spot hybridization using antibody through methyl-cytosine showed a difference in signals. We think that it was due to block of demethylating action of TETs (Bobcat339), and concomitant activation of DNMT1 and DNMT-3B by IL-1b administration (see Figure 7 of the manuscript), showing an increase in methyl-cytosine in co-treatment respect to only bobcat treatment. We attach the figure of spot hybridization in pdf file.
- If AZA is a positive control as mentioned, could this be included in the bone resorption assays
- We added the analysis as requested.

Reviewer 2 Report
1) In the manuscript is required extensive editing of English language and style.
2) In the paragraph (lines 46-51), the objective is poorly written and the same paragraph refers to results, this paragraph should not be in the introduction.
3) In the introduction, the role of IL-1B in epigenetic mechanisms needs to be further elaborated.
4) Correcting text, line 355… as described elsewhere PCR products…
5) In conclusions, it is not correct in the text to refer to another paragraph of the manuscript, in line 448 remove "see Figure 8".
Author Response
Manuscript Number: ijms-2040827
Article Title: The binomial “Inflammation-Epigenetics” in breast cancer progression and bone metastasis: IL-1b actions are influenced by TET inhibitor in MCF-7 cell line.
Journal Name: International Journal of Molecular Sciences
reviewer’s requests
author’s answers
Reviewer #2 Comments
- In the manuscript is required extensive editing of English language and style.
- We thank the reviewer#2 for the suggestion, English language and style has been revised as requested.
- In the paragraph (lines 46-51), the objective is poorly written, and the same paragraph refers to results, this paragraph should not be in the introduction.
- We have modified the text removing results information in introduction and clarified the scope of research.
- In the introduction, the role of IL-1b in epigenetic mechanisms needs to be further elaborated.
- We improved the text reporting information on IL-1b in epigenetic mechanisms.
- Correcting text, line 355… as described elsewhere PCR products…
- We corrected the text.
- In conclusions, it is not correct in the text to refer to another paragraph of the manuscript, in line 448 remove "see Figure 8".
- We thank reviewer #2 for the comments; however, we did not refer to another paragraph but to the next figure (figure 8) concerning the schematic representation of the epigenetic actions of IL-1b and the bobcat 339 block. We probably did not express this clearly and therefore we improved the text trying to make it more understandable.

Round 2
Reviewer 1 Report
Thank you for the additional data, and the changes made.